# Ion-Channel-Targeting Drugs for Chikungunya Virus

**DOI:** 10.3390/molecules30193942

**Published:** 2025-10-01

**Authors:** Hiya Lahiri, Kingshuk Basu, Isaiah T. Arkin

**Affiliations:** 1Department of Biological Chemistry, The Alexander Silberman Institute of Life Sciences, The Hebrew University of Jerusalem, Edmond J. Safra Campus, Jerusalem 9190400, Israel; 2Department of Biomedical Engineering, City University of Hong Kong, Tat Chee Avenue, Kowloon, Hong Kong, China

**Keywords:** chikungunya virus, 6K viroporin, ion channel blockers, antiviral drugs

## Abstract

Alphaviruses are transmitted by Aedes mosquitoes and cause large-scale epidemics worldwide. Chikungunya virus (CHIKV) infection can cause febrile seizures known as chikungunya fever (CHIKF), which ultimately leads to severe joint pain and myalgia. While a vaccine has recently been introduced against CHIKV, at present, no anti-viral drug is available. CHIKV, like other alphaviruses, has a short 6K protein capable of forming an ion channel. Blocking this ion channel with drugs can therefore serve as a potential way to curtail CHIKV infection. To that end, we screened a repurposed drug library using three bacteria-based channel assays to detect blockers against 6K viroporin, yielding several hits. Interestingly, several of the blockers were able to inhibit the 6K protein from the similar Eastern equine encephalitis virus (EEEV), while others were not, pointing to structural specificity which may be explained by modeling studies. In conclusion, our study provides a starting point for developing a new route to potentially inhibit CHIKV.

## 1. Introduction

Chikungunya virus (CHIKV) is a single-stranded positive-sense RNA virus that belongs to the genus *Alphaviruses* and *Togaviridae* family, and is mainly transmitted by *Aedes* mosquitoes [1]. The first case of chikungunya was reported in the Makonde plateau (Tanzania) [2] in 1952 and named in the local language, meaning “that which bends you up”. This describes the posture of the infected patient and how the infection causes severe joint pain. CHIKV outbreaks were reported even before 1779, but those were incorrectly documented as dengue fever. After the 1960s, frequent outbreaks were reported in southeast Asia [3,4,5,6]. In India, major outbreaks were observed in 1973, 2005, and 2010, and the most pronounced was in 2016 [7,8,9,10,11]. Recently, outbreaks have also been observed in the South Pacific region and America [12,13,14].

CHIKV has the same structural and genomic organization as other alphaviruses. It has a lipid bilayer envelope and nucleocapsid shell along with a single-stranded RNA genome of approximately 11.8 kb length [15]. The genome consists of two open reading frames, the first of which encodes a polyprotein that is a precursor of nonstructural proteins nsP1-4, and the second encodes capsid, envelope1 (E1), envelope2 (E2), and two small proteins, 6K and E3. Melton et al. reported that this small 6K protein of alphaviruses can form ion channels [16]. Banerjee and co-workers demonstrated that amantadine can inhibit the 6K ion channel of CHIKV [17]. It also plays a crucial role in viral replication and assembly and has significant importance in the virion release of alphaviruses, confirmed by mutation studies [18,19]. The small cysteine-rich protein forms a cation-selective ion channel in the lipid bilayer and is found to have selectivity for both monovalent and bivalent metal ions (Na^+^, K^+^, and Ca^2+^) [20,21,22], making it an interesting target for blocking.

Many viruses are known to have such ion channels, collectively termed viroporins. One such well-characterized viroporin is the influenza A M2 channel, which can be targeted by aminoadamantane drugs (amantadine and rimantadine) [23,24]. Several other viral ion channels have been found to be successfully targeted by blockers [25,26,27,28,29,30,31,32,33,34].

A vaccine for CHIKV received approval from the FDA in 2023 [35]. A number of studies showed the anti-viral effect of certain compounds by searching for the effects of repurposing drugs [36,37]. Few drugs act by targeting entry inhibition of the virus [38,39]. RNA interference (RNAi) was also used to inhibit viral genome replication and translation [40,41]. Varghese et al. showed the anti-viral activity of abamectin, ivermectin, and berberine against CHIKV infection [42]. However, until now, there has been no specific anti-viral drug available on the market to treat CHIKV infection. This prompted us to search for potential anti-viral agents by targeting the 6K ion channel.

Herein, we targeted 6K viroporin from CHIKV to determine possible anti-viral drugs using bacteria-based channel assays. We screened a repurposed drug library and found nine possible blockers that can serve as the starting point of anti-viral drugs against CHIKV. Moreover, we tested the hit compounds on the 6K ion channel of another *Alphavirus*, Eastern equine encephalitis virus (EEEV). Further, we conducted computational studies to determine the probable structure of the 6K ion channel, which could ultimately increase the specificity of targeting specific viroporin with drugs.

## 2. Results and Discussion

### 2.1. Channel-Based Activity in a Bacterial Setting

To characterize the CHIKV 6K protein as a potential viroporin, and subsequently search for blockers that inhibit its conductivity, we utilized bacteria-based assays. In these assays, channel activity changes the bacterial phenotype and, as such, can point to a protein as a potential viroporin. Subsequently, a search for blockers can be conducted by identifying agents that reverse the channel-driven phenotype.

To ensure proper incorporation within the bacterial inner membrane, we expressed the viral protein as a chimera fused to the carboxy-terminus of the maltose-binding protein. Subsequently, their viroporin activities were examined by three bacteria-based assays, which are detailed in the next sections. Moreover, for comparative purposes, we expressed and examined the 6K protein from another *Alphavirus* genus member—EEEV (sequence conservation and comparison are shown in Appendix A). Finally, the expression of both viral proteins was confirmed by Western blotting, as shown in Appendix A. The presence of both bands near 50 kDa validates the expression of both proteins, and the bands at around 43 kDa signify the presence of native MBP expression in bacterial cells.

#### 2.1.1. Negative Assay

When a channel is overexpressed in the bacterial inner membrane, it results in excessive permeabilization and consequently impairs the ability of the bacteria to grow [28]. Hence, in this assay, the channel eponymously impacts the bacteria negatively. In practice, we examined the outcome of different concentrations of Isopropyl-β-D-1-thiogalactopyranoside(IPTG), which is used to induce expression, on bacterial growth by monitoring optical density (OD) at 600 nm versus time. Figure 1a shows that increasing the inducer concentration from 20 to 200 μM reduces the bacterial growth gradually. For example, 80 μM of IPTG more than halves the growth rate. However, IPTG does not impact the growth rate of bacteria with an empty pMAL vector, as shown in earlier reports [26,43]. Note that not only does the maximum growth rate decrease upon increasing the inducer concentration, but the culture density is also lowered. However, in our experience, growth rate is a more reliable metric for channel activity within a bacterial setting [26,28,30,31,32,33,43,44].

#### 2.1.2. Positive Assay

This assay is reciprocal to the negative assay mentioned above: K^+^–uptake deficient bacteria are incapable of growing in a regular medium unless it is supplemented with high K^+^ concentration [45,46]. However, if the bacteria express a channel capable of potassium transport, growth can be restored, and therefore the channel has a positive impact on its host. Encouragingly, results shown in Figure 1b indicate that bacterial growth is observed upon the induction of CHIKV 6K protein with different concentrations of IPTG. Note that in this assay, lower concentrations of the inducer are used, since at high expression levels, the protein is once more deleterious to growth due to excessive permeabilization (observed in the negative assay).

#### 2.1.3. Fluorescence-pH Assay

In the final experimental assay, a change in the H^+^ intake of bacterial cells on the expression of the viral ion channel was monitored. In this assay, the bacterial cells express a pH-sensitive green fluorescence protein, which has two excitation maxima at 390 and 466 nm, and their ratio is a function of pH [47]. Consequently, a H^+^ influx into the bacteria through a channel triggered by acidification of the media can be readily observed [48]. The result shown in Figure 1c demonstrate that upon increasing the inducer concentration, a corresponding intake of H^+^ is observed.

In conclusion, the above studies have shown that the protein exhibits channel activity in all three bacteria-based assays, and as such can be designated as a viroporin. The 6K from CHIKV was able to reduce the bacterial growth when the inducer concentration increased gradually in the negative assay (Figure 1a). Channel activity was also confirmed for CHIKV 6K by the positive assay, where protein expression increases the growth of K^+^–uptake-deficient bacteria at low K^+^ concentration (Figure 1b). Finally, the 6K protein was able to increase the H^+^ flow into the cell, which was detected by a chromosomally encoded GFP fluorescence change (Figure 1c). Taken together, all bacteria-based assays were able to demonstrate channel activity of the 6K protein from CHIKV.

### 2.2. Drug Screening

After having shown in three independent bacteria-based assays that CHIKV 6K protein exhibits viroporin characteristics, we set forth to search for blockers that can inhibit its channel activity. We adapted a screening method by complementary assays based on bacterial genetic selection. The channel expression and the activity of targeted blockers were quantified as a function of bacterial growth. This makes drug screening easier and less time-consuming, and produces fewer false positive results, especially while screening sizable drug libraries. To that end, a repurposed drug library of 2839 compounds was first screened on the 6K channel from CHIKV using the negative assay. Here, active compounds, upon addition to the screening assay, should increase the bacterial growth since they reverse the negative impact of the channel on its heterologous host. The active compounds were then subjected to the positive assay, where they are expected to hinder bacterial growth, since, in this instance, the viroporin is beneficial to its host. The reciprocal nature of both positive and negative assays decreased erroneous results. Finally, the hit compounds were checked by the fluorescence–pH assay to further minimize false positive results.

From Figure 2a, it can be seen that nine compounds exhibit activity in the negative assay by increasing the bacterial growth rate: sulfabenzamide, tarenflurbil, sodium phenylbutyrate, tocofersolan, 5-azacytidine, pentamidine, arterolane, saroglitazar, and plerixafor. Among them, sulfabenzamide and arterolane are capable of restoring bacterial growth to levels that approach values that are seen when the bacteria do not harbor any channel. These nine compounds are also active in the positive assay, where a reduction in bacterial growth is the indicator of activity (Figure 2b). In addition, all nine blockers scored positively in the fluorescence–pH assay. Figure 2c shows that the said nine compounds resulted in a reduction in the fluorescence intensity, once more providing proof of their activity.

Finally, a dose–response analysis of the said nine compounds was conducted, employing the negative and positive assays as shown in Figure 3. The data from the negative assay were used to calculate Monod coefficients (*K*_*s*_) for each drug, as shown in Appendix A.

### 2.3. Comparison with 6K EEEV

Another important member of the *Alphavirus* genus is the EEEV. A comparison of the 6K protein sequences from CHIKV and EEEV revealed appreciable sequence identity between the two 6K proteins of 52% (Figure 4), in particular in the transmembrane (TM) region (Figure 4). Moreover, the sequence conservation study reveals higher sequence conservation in the TM domain of both channels (Appendix A). Since we have previously shown that EEEV 6K protein exhibits ion channel activity [26], we sought to compare the activity of both proteins. Such a comparison may provide additional insights into the detailed structure and function of both proteins. To that end, we utilized the positive and negative bacteria-based assays to demonstrate the channel activity once more. Indeed, the results shown in Appendix A depict similar activity in the negative and positive assays.

Considering that both proteins exhibit similar channel characteristics in the bacteria-based assays, we decided to examine the effects of the blockers that inhibited the 6K viroporin from CHIKV on the 6K channel from EEEV. Interestingly, as shown in Figure 5, it was observed that the nine hit compounds that were active against CHIKV 6K exhibited varying activities against the EEEV protein. In particular, arterolane was again the most potent blocker against both proteins. However, 5-azacytidine, sulfabenzamide, and saroglitazar, which were the second, third, and fourth most active compounds against CHIKV 6K, respectively, exhibited little to no activity in either assay against EEEV 6K. The rest of the compounds exhibited different activity ranges, as tabulated in Table 1.

To avoid the probability of non-specificity, we checked the three hit compounds on an empty vector and also on the M2 channel (well-characterized channel) from influenza, and no activity was observed.

The differences and similarities in the activities between the various hits can be attributed to the sequences of the two proteins. For example, the binding site of arterolane, tarenflurbil, and tocofersolan (active against both proteins) is likely unchanged between the two proteins. However, the same cannot be said of the binding site for sulfabenzamide, which exhibits activity only against the 6K protein from CHIKV.

### 2.4. Structural Analyses

In order to gain a more detailed understanding of the 6K channels from the CHIKV and EEEV, we proceeded with structural analyses. Since, to the best of our knowledge, the structures of either protein have not been determined experimentally, we proceeded by employing AlphaFold2 [50] to predict the structural features from the corresponding amino acid sequences.

One important consideration in utilizing the above strategy is the oligomeric state of the protein. In particular, due to their small size, many viroporins form a homo-oligomeric symmetric complex surrounding a pore [51,52]. Therefore, as an input to AlphaFold2, we utilized the primary sequences of the proteins at different degrees of oligomerization, from dimer to hexamer.

In order to select structures for detailed investigation, two criteria were enacted in series: (i) The structure must have a pore. Subsequently, (ii) structures with a pore are sorted according to their value in the site-wise confidence score provided by AlphaFold2 [50]. Supporting Appendix A show the Local Difference Distance Test (lDDT) plots for different oligomers. The residue-wise lDDT-score is a superposition-free score representing the local atomic-distance difference. This score provides a residue-wise quality factor for the predicted structure [53]. For 6K from the CHIKV, the trimer produces the highest score with a porous structure, whereas the tetramer and pentamers also possess pores but have lower confidence values. On the other hand, for the 6K viroporin from EEEV, though the tetramer and hexamer have higher confidence scores than the trimer, they do not form pore-like structures. Hence, in both instances, a trimeric assembly was utilized for future studies.

Molecular dynamics simulations were subsequently conducted on the protein structures for 300 ns by incorporating them within an ERGIC-like (ER-Golgi intermediate compartment) membrane to mimic the native conditions. Figure 6a shows the root mean square deviation of both proteins during the simulations as a function of time. It can be seen that, for both membrane–protein assemblies, convergence is reached after 150 ns. To further compare the dynamics of the two proteins in a membrane environment, sequence-wise root mean square fluctuations (rmsf) were measured for each protein and plotted in Figure 6b. Blue and orange shaded areas mark the transmembrane domains for 6K CHIKV and 6K EEEV, respectively. It is evident from Figure 6b that both proteins show similar dynamic behavior of the transmembrane domain, with variations in fluctuation of the outer membrane segments.

Such stable structures with RMSD values for each protein were overlayed (approximately 40 structures for each protein) to yield a representative structure of the proteins within the lipid environment, as shown in Figure 7. In both cases, though the oligomerization mode is the same, EEEV 6K exhibits a different channel geometry. This can be due to the presence of higher fluctuations in the post-TM region (from Leu40 onwards) compared to the CHIKV counterpart. Moreover, the time-averaged Ramachandran plot [54] of 40th to 46th amino acid residues (Appendix A) of both proteins reveals a break in the helical geometry of 6K from EEEV. This deviation is also visible in the molecular model, shown in Figure 7b, inset. Such factors may have a significant effect on the dissimilarities in drug interaction with the viroporins (Table 1). To elucidate the pore shape of 6K proteins, we utilized PoreWalker [55] to analyse pore geometry. Appendix A shows pore geometry analysis for 6K CHIKV and EEEV. It is evident from the figure that the pore geometry of CHIKV 6K is very different from the EEEV counterpart. The CHIKV 6K shows an open, funnel-like geometry (summed up as SUS; see figure for details), whereas EEEV 6K shows a bottle-like geometry with narrow ends (summed up as USDS; see figure for details). These different pore geometries can be a reason for the varied activity of the same drugs in both ion channels.

## 3. Materials and Methods

### 3.1. Protein Sequence

The sequences of the CHIKV and EEEV 6K proteins were obtained from the NCBI databases GenBank:WGZ84057.1 [56] and GenBank:AMT80038.1 [57], respectively. Potential transmembrane domains of both proteins were identified using TMHMM [58]. The conserved sequences were displayed in LOGO format using the *Seq2Logo* web server [59].

### 3.2. Cloning

gBlocks of the aforementioned gene sequences were constructed with the desired sequence along with a histidine tag and stop codon at the C-terminal end and ordered from Integrated DNA Technologies (Coralville, IA, USA). All the proteins were then expressed as a chimera with the maltose-binding protein using the pMAL-p2X plasmid (New England BioLabs; Ipswich, MA, USA), employing a Gibson assembly (Gibson Assembly^®^ Cloning Kit from New England BioLabs).

### 3.3. Bacterial Strain

Different strains of K12 *Escherichia coli* were used for this study: DH10B (Invitrogen; Carlsbad, CA, USA); LB650 [45] potassium uptake deficit strain (Δ*trkG*, Δ*trkH*, and Δ*kdpABC5*), which were a kind gift from Professor K. Jung, Ludwig-Maximilians Universität München and Professor G.A. Berkowitz, University of Connecticut; and the LR1 strain, containing a chromosomal copy of a pH-sensitive green fluorescent protein [47], which was kindly donated by Professor M. Willemoës and Professor K. Lindorff-Larsen (Københavns Universitet).

### 3.4. Chemicals

Isopropyl-β-D-1-thiogalactopyranoside (IPTG) was purchased from Biochemika-Fluka (Buchs, Switzerland). All other chemicals were purchased from Sigma-Aldrich Laboratories (Rehovot, Israel).

### 3.5. Bacteria Based Assays

Three bacteria-based channel assays were used to determine the channel activity of the above-mentioned proteins.

#### 3.5.1. Negative Assays

E. coli DH10B either containing the viral chimera or lacking it (as a control group) was grown overnight in Lysogeny Broth (LB) at 37 °C. Then, on the next day, the culture was diluted 500-fold as a secondary culture and subjected to further growth until it reached an OD_600_ of 0.2. Subsequently, in a 96-well plate containing 50 μL of the desired solution, another 50 μL of the bacterial culture was added, and control wells were prepared without IPTG. Different concentrations of IPTG were added for the induction, alongside 1% D-glucose. The 96-well plates were incubated in a multi-plate reader (LogPhase 600 from BioTek; Santa Clara, CA, USA) at 37 °C for 16 h. Readings were taken at 15 min time intervals and plotted with respect to time.

#### 3.5.2. Positive Assays

The positive assay employed the same protocol as the negative assay described above, with the following differences: the LB650 bacterial strain was used instead of DH10B, and LB media was replaced with LBK media containing 150 mM KCl instead of NaCl.

#### 3.5.3. Fluorescence pH (Acidity) Assay

For this assay, a special type of bacterial strain (LR1) was used that contains a chromosomal copy of a pH-sensitive green fluorescent protein [48]. These LR1 cells with different viroporin chimeras were grown overnight in LB media containing 1% D-glucose. The next day, the culture was diluted 500-fold and grown until the OD_600_ reached 0.6 to 0.8. Induction was conducted with different concentrations of IPTG and diluted to reach an OD_600_ of 0.2. Cultures without IPTG were used as control. After an OD_600_ of 0.2 was reached, cells were pelleted down by centrifuging at 3000 g for 10 min. The cells were then washed and dissolved in McIlvaine Buffer [60] (200 mM Na2HPO4 and 0.9% NaCl adjusted to pH 7.6 with 0.1 M citric acid). Subsequently, 200 μL of cell culture was added to each well containing 30 μL of McIlvaine buffer in a 96-well plate (Nunclon f96 Microwell Black Polystyrene, Thermo Fisher Scientific; Waltham, MA, USA). Three wells contained McIlvaine buffer and bacterial cultures, but without induction as a control. Then, 70 μL of citric acid (300 mM, 0.9% NaCl) was added at the starting point to each well by a liquid handling system (Tecan; Männedorf, Switzerland). The fluorescent measurements were carried out at an ambient temperature in a microplate reader (Infinite F200 Pro, Tecan Group), setting the emission at 520 nm and alternating between 390 and 466 nm excitations. The proton concentrations were then calculated using the ratio of two wavelengths according to [48].

#### 3.5.4. Drug Screening Assay

A chemical library consisting of 2839 compounds purchased from MedChem Express (HY-L035, Monmouth Junction, NJ, USA) was subjected to bacteria-based screening on corresponding channels. According to the vendor, “the library contains approved drugs and compounds that have passed phase I clinical trials, which have been completed extensive pre-clinical and clinical studies and have well-characterized bio-activities, safety, and bio-availability properties”.

All the compounds were used at a concentration of 100 μM. Initial screening was performed with the negative assay. Subsequently, compounds that passed through the negative assay by increasing bacterial growth were subjected to the positive assay, which is exactly reciprocal to the previous one. The compounds that showed activity in both assays were then assessed with the acidity assay. Initial screening was performed on the CHIKV 6K channel, and hit compounds were tested on another 6K channel from EEEV.

### 3.6. Western Blot

The CHIKV and EEEV 6K-maltose binding protein (6K-MBP) chimera was expressed in DH10B cells at 26 °C overnight after induction with 0.1 mM IPTG. Cells were then calibrated for OD_600_ of 1. The over-expressed cells were pelleted at 5000× *g* for 5 min at 4 °C. The cell pellets were lysed using lysis buffer (50 mM Tris pH 8.0, 10% Glycerol, 1% Triton X-100, 2 mM PMSF, 100 µg/mL DNAse, and 100 µg/mL lysozyme). The cells were then kept at 37 °C for 15 min on a dry bath and then freeze-thawed twice, immediately followed by ultrasonication for three cycles of 30 s with a 10 s gap at 40 watts (Vibra-Cell™ Ultrasonic Liquid Processors; Newtown, CT, USA). Cell lysates were collected after centrifugation (15,000× *g* for 20 min, 4 °C) and treated with 5X sample buffer solution (5X solution of 250 mM Tris·HCl, pH 6.8, 10% SDS, 30% (*v*/*v*) Glycerol, 10 mM DTT, 0.05% (*w*/*v*) Bromophenol Blue) in a 5:1 ratio followed by heating for 5 min in a dry bath at 100 °C. The lysate was then resolved using electrophoresis, followed by the transfer of the antigens to nitrocellulose membranes using a Semi-dry Trans-Blot^®^ Turbo™ Transfer System (BIO-RAD; Hercules, CA, USA). The membranes were blocked with TBS-T (tris-buffered saline containing Tween-20) containing 1% low-fat milk, incubated overnight with a primary antibody (Anti-MBP, New England BioLabs), washed three times with TBS-T, incubated for two hours with a secondary antibody linked to horseradish peroxidase (anti-rabbit, Novagen; Temecula, CA, USA), and washed once again with TBS-T. Immunoreactive bands were detected using an ECL kit (Biological Industries; Haemek, Israel).

### 3.7. Dissociation Constant Calculation

The Monod coefficients (*K_s_*) for each hit compound were calculated by using Prism 10 (GraphPad Software; San Diego, CA, USA). The data obtained are non-linearly fit, and the data are given in the Appendix A.

### 3.8. Computational Studies

Primary protein sequences (oligomeric) were input into AlphaFold2.ipynb using ColabFold v1.5.2: AlphaFold2 to obtain tentative three-dimensional structures [50]. The proteins were subsequently embedded within a pre-equilibrated ER–Golgi intermediate compartment (ERGIC) membrane composed of 1-palmitoyl-2-oleoyl-sn-glycero-3-phosphoethanolamine, 1-palmitoyl-2-oleoyl-sn-glycero-3-phosphocholine, and 1-palmitoyl-2-oleoyl-sn-glycero-3-phospho-L-serine at a 3:1:1 ratio using CHARMM-GUI [61,62,63]. Subsequently, position-restraints (100,000 kcal/mol/Å^2^) on protein-heavy atoms were used to ensure that the protein does not change during energy minimization, accomplished using the steepest descent minimization algorithm with a tolerance of 500 kj mol^−1^ nm^−1^.

Molecular dynamics (MD) simulations for all the systems were carried out for 200 ns using GROMACS version 2022.3 [64,65,66,67,68] and CHARMM36m force field [69]. A LINCS algorithm with integration time step of 2 fs was used in all cases to constrain the length and angles of hydrogen atoms [70]. Atomic coordinates were saved at every 500 ps interval. The temperature for reference was set at 323 K, and solvents, lipids, and proteins were coupled separately to a Nosé–Hoover temperature bath [71,72] with a coupling constant value τ = 0.5 ps. Pressure coupling was obtained with a Parrinello–Rahman barostat with τ = 2 ps [73,74]. A 1.2 nm cut-off was set for van der Waals interactions. The updating frequency of the neighbor list was kept at 10 fs. Electrostatic parameters were calculated using 4th order Particle Mesh Ewald (PME) long-range electrostatics [75] and the cut-off for short-range electrostatics was set to 1.2 nm.

All the simulation boxes contained 190 lipid molecules and approximately 19,481 water molecules, with Na^+^ and Cl^−^ counter ions (at 0.15 M concentration). Water molecules were fitted in the FLEXSPC model [76] and the total number of atoms was in the 86,000 range. RMSD values versus time were extracted, and structures of the simulated proteins were visualized and analyzed with VMD software version 1.9.4a51 [77].

## 4. Conclusions

The treatment of CHIKV infection is challenging due to the lack of anti-viral drugs. However, the severity of CHIKV infection needs immediate attention to control the epidemic. In this study, we have screened a library of repurposed drugs using bacteria-based ion-channel blocking experiments. We found nine effective drugs against the chikungunya virus 6K ion channel and checked them on another 6K variant from Eastern equine encephalitis virus. Interestingly, three of the compounds showed significant activity. We concluded by employing structure prediction tools and molecular dynamics simulation to elucidate the probable structures of the corresponding ion channels within the lipid bilayer. This study can be further extended to in vitro and in vivo assays [78] to confirm the drug activity. Rigorous electrophysiological experiments can be performed to characterize binding events at the molecular level.

## Figures and Tables

**Figure 1 molecules-30-03942-f001:**
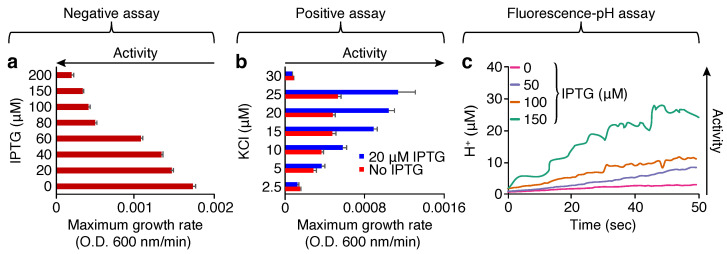
Three bacteria-based channel assays to assess the ion channel activity of chikungunya virus 6K protein: (**a**) negative assay, (**b**) positive assay, and (**c**) fluorescence-pH assay.

**Figure 2 molecules-30-03942-f002:**
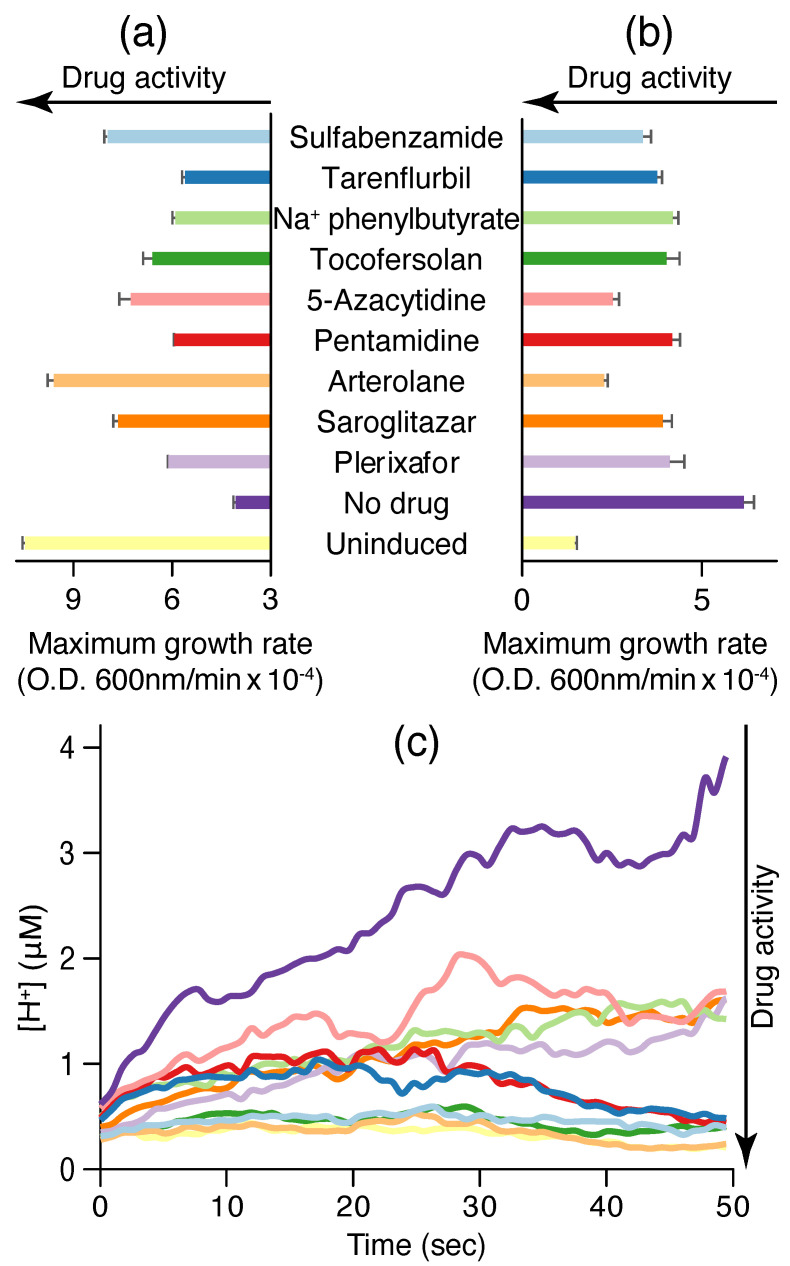
Blocker screening results against CHIKV 6K viroporin using the (**a**) negative assay, (**b**) positive assay, and (**c**) acidity assay. Drug activity in each of the assays is denoted by arrows. Bacteria in which the channel expression was not induced serve as a control group. Note reciprocal activity of the drugs in the negative and positive assays. Blocker concentration was 100 μM.

**Figure 3 molecules-30-03942-f003:**
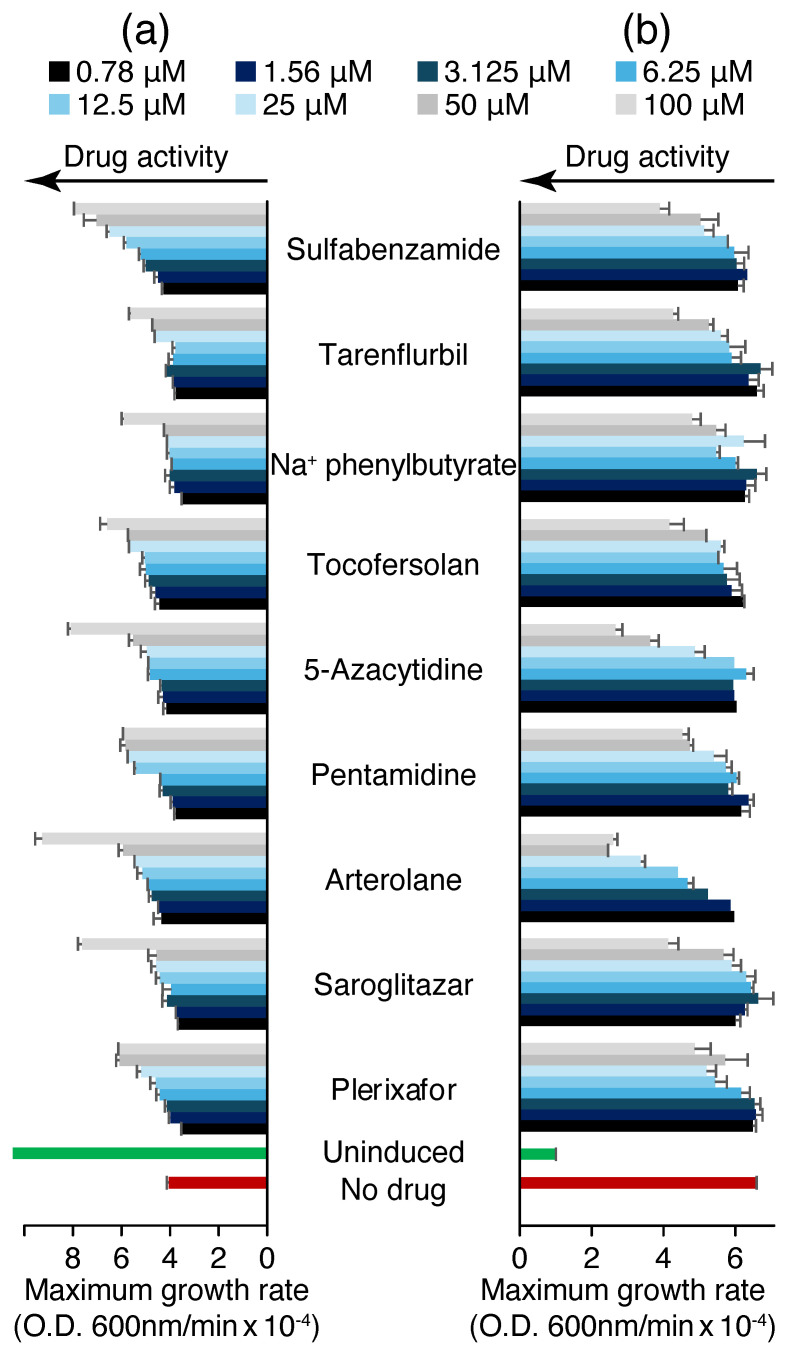
Blocker screening results against 6K CHIKV viroporin utilizing (**a**) the negative and (**b**) the positive assays. Maximum growth rates are shown as a function of different compound concentration (0.78–100 µM). Bacteria in which channel expression was not induced (in red) are used as the control. Drug activity in each of the assays is denoted by arrows.

**Figure 4 molecules-30-03942-f004:**
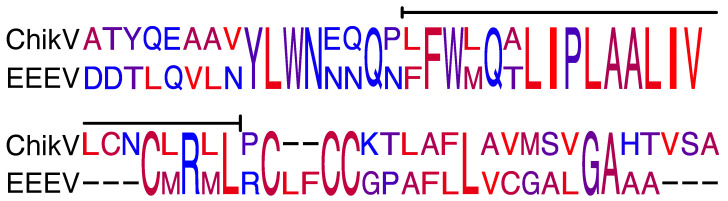
Amino acid sequences and conservation of 6K proteins of CHIKV (**top**) and EEEV (**bottom**). The color-coding is according to hydrophobicity, whereby red is the most hydrophobic and blue is hydrophilic. The bar represents the potential transmembrane domain according to TMHMM [49].

**Figure 5 molecules-30-03942-f005:**
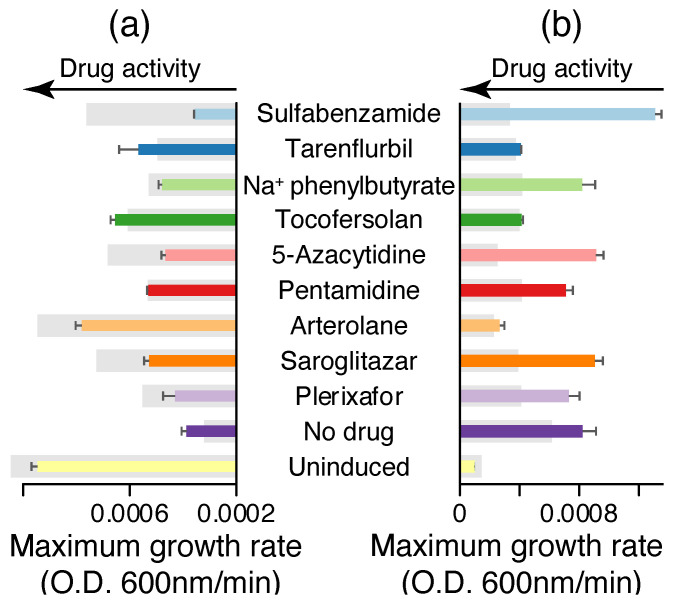
Activity of hit compounds obtained from 6K CHIKV blocker screening on the 6K EEEV by (**a**) the negative assay and (**b**) the positive assay. The gray rectangles indicate the corresponding activity obtained against the CHIKV 6K viroporin shown in Figure 2. Drug activity in each of the assays is denoted by arrows.

**Figure 6 molecules-30-03942-f006:**
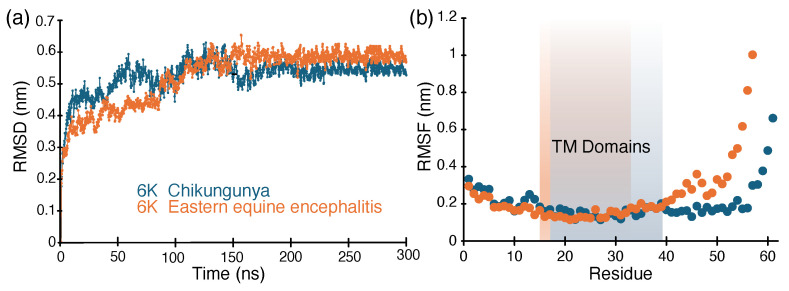
(**a**) Root mean square deviation as a function of time plots of protein–membrane assemblies (blue: CHIKV 6K and orange: EEEV 6K protein trimers). (**b**) Residue-wise root mean square fluctuation (averaged from 250 to 300 ns) (orange: CHIKV 6K and blue: EEEV 6K protein trimers). Transmembrane domains are highlighted in blue and orange for CHIKV 6K and EEEV 6K, respectively.

**Figure 7 molecules-30-03942-f007:**
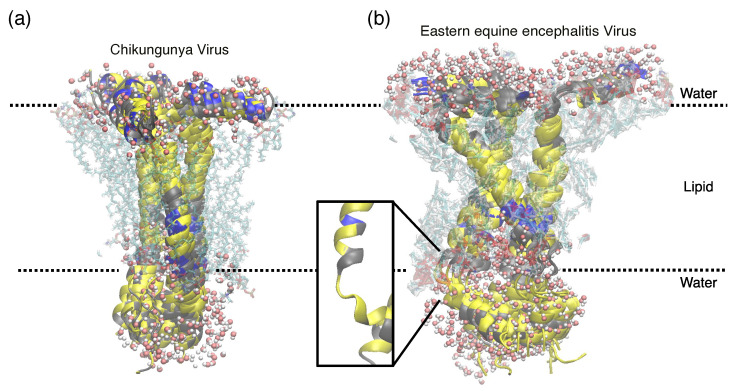
MD simulated overlayed ensemble structures of 6K in (**a**) CHIKV and (**b**) EEEV embedded within a lipid bilayer. Rough bilayer boundaries are shown in dotted lines with lipid and water molecules within 5 Å vicinity of proteins (color coding: hydrophobic residues in yellow; charged residues in blue).

**Table 1 molecules-30-03942-t001:** Comparison of drug screening results between CHIKV and EEEV 6K proteins. Raw data are found in Figure 2 and Figure 5 and were calculated by comparing the growth rates of uninduced and untreated bacteria to the growth rates of drug-treated bacteria. The average activity is the mean of the results of the negative and positive assays.

	CHIKV			EEEV		
	Neg.	Pos.	Ave.	Neg.	Pos.	Ave.
Sulfabenzamide	61%	60%	60%	−5%	−67%	−36%
Tarenflurbil	24%	51%	38%	32%	57%	45%
Na+ phenylbutyrate	29%	42%	35%	16%	0%	8%
Tocofersolan	39%	46%	43%	48%	57%	52%
5-Azacytidine	50%	78%	64%	14%	−13%	1%
Pentamidine	29%	42%	36%	25%	15%	20%
Arterolane	86%	83%	84%	70%	77%	74%
Saroglitazar	56%	48%	52%	25%	−12%	7%

## Data Availability

All data are available upon request from the corresponding authors.

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
