# Peer review of "Ion-Channel-Targeting Drugs for Chikungunya Virus"

_molecules, 2025, doi:10.3390/molecules30193942_

Round 1

Reviewer 1 Report

Comments and Suggestions for Authors

The authors aim to identify drugs for treating viruses through the 6K protein. Using three sets of screening methods, they have obtained several potential drugs. However, the experimental results are quite preliminary, and the detection method cannot prove that these drugs act on the virus. The main issues are as follows:

  1. It is necessary to rule out the influence of IPTG and add groups with no plasmids and back bone as controls.
  2. There is no background of the 6K protein at all in the introduction, making it impossible to understand why the authors selected this protein as a target.
  3. The protein structure content in Section 3.3.1 is irrelevant to the main theme of this paper, and including this part is redundant.
  4. Concluding that the pathway is inhibited is not feasible only by measuring bacterial growth without corresponding results of the 6K protein expression level under the same conditions.

Reviewer 2 Report

Comments and Suggestions for Authors

The manuscript “Ion channel targeting drugs for CHIKV” by Lahiri et al presents a compelling and well-executed drug repurposing screen targeting the CHIKV 6K viroporin. Overall the study is comprehensive, structured and provide a major step in our understanding of the potential drug candidates against CHIKV and establishes the viroporin a target for drug screening. But few important experimental evidences are missing in the results to establish the hit compounds to be antivirals and should be addressed before accepting it for publication.

  1. Authors use bacterial screening system to identify potential inhibitors of CHIKV 6K Viriporins. However, to establish these compounds as antiviral candidates, essential validation is required in biological context. Author should determine antiviral efficacy (EC50) in mammalian cells, alongside cytotoxicity (CC50) and selectivity index CC50/EC50. Without these critical data, this study remains in vitro channel blockade screen but can not claim compounds as a new antiviral candidate against CHIKV and lack any therapeutic potential.
  2. Initial screening experiments demonstrate functional blockade, but does not established the direct binding of compound to the channel pore or any other mechanism (allosteric effect). To provide direct evidence authors can perform electrophysiology experiments on purified protein or if it is beyond the scope, this limitation should be explicitly acknowledged and noted as future goal.
  3. In Fig 6a; authors recorded convergence after 150 ns, but it looks system is not stabilized in CHIKV 6K. Authors should extend the run up to 300 ns to confirm the convergence.

Round 2

Reviewer 2 Report

Comments and Suggestions for Authors

Authors have addresses all the concerns. I recommend manuscript for publication